# Soybean (*Glycine max* L.) Leaf Moisture Estimation Based on Multisource Unmanned Aerial Vehicle Image Feature Fusion

**DOI:** 10.3390/plants13111498

**Published:** 2024-05-29

**Authors:** Wanli Yang, Zhijun Li, Guofu Chen, Shihao Cui, Yue Wu, Xiaochi Liu, Wen Meng, Yucheng Liu, Jinyao He, Danmao Liu, Yifan Zhou, Zijun Tang, Youzhen Xiang, Fucang Zhang

**Affiliations:** 1Key Laboratory of Agricultural Soil and Water Engineering in Arid and Semiarid Areas of Ministry of Education, Northwest A&F University, Xianyang 712100, China; 2022012387@nwsuaf.edu.cn (W.Y.); 2022012367@nwsuaf.edu.cn (G.C.); 2022012353@nwsuaf.edu.cn (S.C.); 2022012363@nwsuaf.edu.cn (Y.W.); 2023055903@nwafu.edu.cn (X.L.); mengwen@nwsuaf.edu.cn (W.M.); 2022012268@nwsuaf.edu.cn (Y.L.); 2022012263@nwsuaf.edu.cn (J.H.); 2022012351@nwsuaf.edu.cn (D.L.); 2022012368@nwsuaf.edu.cn (Y.Z.); tangzijun@nwsuaf.edu.cn (Z.T.); youzhenxiang@nwsuaf.edu.cn (Y.X.); zhangfc@nwsuaf.edu.cn (F.Z.); 2Institute of Water-Saving Agriculture in Arid Areas of China, Northwest A&F University, Xianyang 712100, China

**Keywords:** leaf moisture content, multispectral, soil moisture content, soybean, texture features, vegetation indices

## Abstract

Efficient acquisition of crop leaf moisture information holds significant importance for agricultural production. This information provides farmers with accurate data foundations, enabling them to implement timely and effective irrigation management strategies, thereby maximizing crop growth efficiency and yield. In this study, unmanned aerial vehicle (UAV) multispectral technology was employed. Through two consecutive years of field experiments (2021–2022), soybean (*Glycine max* L.) leaf moisture data and corresponding UAV multispectral images were collected. Vegetation indices, canopy texture features, and randomly extracted texture indices in combination, which exhibited strong correlations with previous studies and crop parameters, were established. By analyzing the correlation between these parameters and soybean leaf moisture, parameters with significantly correlated coefficients (*p* < 0.05) were selected as input variables for the model (combination 1: vegetation indices; combination 2: texture features; combination 3: randomly extracted texture indices in combination; combination 4: combination of vegetation indices, texture features, and randomly extracted texture indices). Subsequently, extreme learning machine (ELM), extreme gradient boosting (XGBoost), and back propagation neural network (BPNN) were utilized to model the leaf moisture content. The results indicated that most vegetation indices exhibited higher correlation coefficients with soybean leaf moisture compared with texture features, while randomly extracted texture indices could enhance the correlation with soybean leaf moisture to some extent. RDTI, the random combination texture index, showed the highest correlation coefficient with leaf moisture at 0.683, with the texture combination being Variance1 and Correlation5. When combination 4 (combination of vegetation indices, texture features, and randomly extracted texture indices) was utilized as the input and the XGBoost model was employed for soybean leaf moisture monitoring, the highest level was achieved in this study. The coefficient of determination (R^2^) of the estimation model validation set reached 0.816, with a root-mean-square error (RMSE) of 1.404 and a mean relative error (MRE) of 1.934%. This study provides a foundation for UAV multispectral monitoring of soybean leaf moisture, offering valuable insights for rapid assessment of crop growth.

## 1. Introduction

Soybean (*Glycine max* L.), as one of the major leguminous crops globally, plays a crucial role in global food security and sustainable agriculture [1]. In arid and semi-arid regions, soybean cultivation faces multiple challenges, often associated with limited water resources and irregular precipitation patterns [2]. Being a water-consuming crop, soybean requires adequate water for normal growth [3]; however, water scarcity in dry areas frequently leads to water stress, constraining soybean growth and resulting in yield reduction [4]. Therefore, timely monitoring of soybean leaf moisture is essential for identifying plant moisture status, adjusting irrigation strategies, and enhancing yield.

Plant leaf moisture is influenced by multiple factors such as sunlight, soil moisture, and air temperature, making it difficult to measure accurately and rapidly [5]. Currently, commonly used methods for measuring plant leaf moisture include oven drying, Karl Fischer titration, and capacitance methods; however, these methods are often time-consuming, labor-intensive, and limited in applicability, failing to provide timely and accurate field monitoring data [6]. Hence, rapid acquisition of plant leaf moisture status and timely adjustment of soil moisture management strategies remain a challenge for large-scale agricultural operations.

Remote sensing technology has been widely applied to qualitatively and quantitatively analyze the water and nutrient status of large-scale plants nondestructively [7]. Among them, multispectral remote sensing technology can simultaneously obtain data from multiple bands, covering various information during the crop growth cycle [8]. In contrast, traditional field measurement methods require measurements at different times and locations, which are inefficient and make it difficult to achieve comprehensive monitoring [9]. Compared with hyperspectral remote sensing, multispectral remote sensing has fewer bands but still covers key bands related to crop growth and health conditions, with more flexible band selection according to specific application requirements, making UAV multispectral technology more widely applicable in field management [10].

Spectra can focus on the internal optical response of crops [11], while images capture external morphological information of crops [12]. Multispectral data provide reflectance of crops in different bands, which correlates with crop leaf moisture to some extent [13], thus enabling indirect inference of crop moisture status through multispectral data analysis. Vegetation indices, computed based on multispectral data, directly reflect the growth status of crops [14]. On the other hand, vegetation canopy texture features, obtained through image data analysis, reflect the spatial distribution and structural characteristics of crops, including leaf morphology, density, and arrangement, which also indicate the moisture status of crops [15]. Researchers have conducted relevant studies on monitoring physiological growth indicators of crops based on vegetation indices; however, constrained by crop types and meteorological factors, using fixed formulas to calculate vegetation indices for monitoring physiological growth indicators of crops can limit prediction accuracy. Some studies have shown that combining texture features with vegetation indices can improve the inversion accuracy of physiological growth indicators of crops (biomass [16], leaf area index [17], chlorophyll content [18], etc.). Constructing inversion models based on multiple input variables has higher accuracy compared with single input variables.

Machine learning methods have been proven effective in solving complex nonlinear problems with multiple factors. Some results indicate that back propagation neural networks (BPNN) have higher accuracy in monitoring physiological growth indicators [19,20], while other studies suggest that extreme gradient boosting (XGBoost) may be more suitable [21,22]. Overall, there is uncertainty in the existing research regarding the optimal feature extraction and modeling methods for monitoring physiological growth indicators. Therefore, this study further explores monitoring soybean leaf moisture.

In this study, our aim was to determine the relationship between soybean leaf moisture and vegetation indices and texture features. To achieve this, we employed machine learning algorithms such as ELM, XGBoost, and BPNN to explore the optimal combination of these features and the best monitoring depth for monitoring soybean physiological growth indicators, aiming to provide rapid and efficient theoretical support for field water management.

## 2. Materials and Methods

### 2.1. Research Area and Test Design

The experiment was conducted during 2021–2022 at the Institute of Water-saving Agriculture in Arid Areas, Northwest A&F University, located in the southern part of the Loess Plateau in northwest China (34°14′ N, 108°10′ E). The experimental area is a typical dryland agricultural region with an average annual precipitation of 632 mm and an evaporation of 1500 mm. Daily temperature and rainfall data for the two growing seasons were carefully recorded by an automatic weather station located within the experimental fields, as shown in Figure 1. During the period from June to October 2021, the annual average maximum and minimum temperatures were 30.3 °C and 20.0 °C, respectively, while in 2022, they were 31.3 °C and 21.2 °C, respectively. The precipitation during the soybean growing seasons in 2021 and 2022 was 432.6 mm (from 18 June 2021 to 30 September 2021) and 279.5 mm (from 10 June 2022 to 20 September 2022), respectively. For basic terrain and meteorological information of the experimental site, please refer to reference [1].

In this experiment, a split-plot design with two factors was employed, including different cover treatments and supplementary irrigation strategies. The cover treatments consisted of three types: straw mulch (SM), ridge-film mulch (FM), and no mulch (NM). Additionally, three supplementary irrigation treatments were included: W1 (irrigation during branching stage, V4), W2 (irrigation during podding stage, R2), and W3 (irrigation during both V4 and R2 stages simultaneously). This resulted in a total of nine treatments, each replicated three times, comprising 27 experimental plots. Each irrigation event applied 40 mm of water. The detailed information of the experiment is presented in Table 1. Each plot had an area of 24 m^2^ (4 m × 6 m) and was arranged randomly, with a 2 m buffer zone around each plot.

Before sowing, each plot received phosphorus and potassium fertilizers at a rate of 30 kg ha^−1^ and nitrogen fertilizer at a rate of 120 kg ha^−1^. The nitrogen fertilizer used in the experiment was urea (46% N), the phosphorus fertilizer was calcium superphosphate (16% P_2_O_5_), and the potassium fertilizer was potassium chloride (62% K_2_O).

For the FM treatment, a ridge (50 cm wide, 30 cm high) was used, and seeds were sown in furrows covered by the ridge. The ridge–furrow ratio was 1:1. Before sowing, two rows of soybeans were planted at the bottom of each ridge. The straw mulch rate was 9000 kg ha^−1^, and wheat straw was used to cover the soil within 7 days after sowing. The planting density of soybeans was 300,000 plants ha^−1^, with row spacing of 50 cm and plant spacing of 10 cm. Soybeans were sown on 18 June 2021 and 10 June 2022 and harvested on 30 September 2021 and 20 September 2022, respectively.

Additionally, to ensure proper germination, approximately 20 mm of water was applied to each plot after sowing. Other field management practices, including spraying and weeding, remained consistent with local practices.

### 2.2. Data Collection and Preprocessing

#### 2.2.1. Drone Data Acquisition

This study utilized a DJI Matrice M300 RTK quadcopter equipped with an MS600 Pro multispectral camera platform to acquire multispectral remote sensing data. The camera platform comprised six spectral channels and was equipped with six CMOS image sensors, with a pixel resolution of 1.2 × 10^6^. The sensors covered the following spectral bands: blue band (center wavelength 450 nm, band 1), green band (center wavelength 555 nm, band 2), red band (center wavelength, band 3), red edge band 1 (center wavelength 720 nm, band 4), red edge band 2 (center wavelength 750 nm, band 5), and near-infrared band (center wavelength 840 nm, band 6). Data were collected during the soybean flowering period (5 August 2021 and 10 August 2022) at noon under clear-sky conditions. Flight routes were planned for the study area, and whiteboard calibration was conducted. The flight altitude was set at 30 m, with a speed of 2.5 m per second and a pixel resolution of 4.09 cm. The forward and lateral overlap ratios were set at 75% and 65%, respectively. Figure 2 shows the aerial photos of some residential areas in the test area.

#### 2.2.2. Obtaining Leaf Moisture Content

Simultaneously with the collection of multispectral information by drones, the soybean leaf moisture content was determined using the drying method. Five average growing soybeans plants were selected from each plot. Fresh leaves were harvested from various directions and heights of each plant, totaling 100 g, using an analytic balance. These leaves were then placed in parchment bags, labeled, and subjected to dehydration in a drying oven at 105 °C for 0.5 h, followed by drying at 80 °C until a constant mass was achieved. The dry mass, after subtracting the mass of the parchment bags, represented the moisture content. The average moisture content of the five soybean plants was considered indicative of the entire plot’s soybean leaf moisture content.

#### 2.2.3. Multispectral Image Processing

In scientific research, precise handling and analysis of remote sensing data are crucial. In this study, we employed the Yusense Map V2.2.2 software to process multispectral imagery collected by unmanned aerial vehicles (UAVs). Initially, the software was used for image mosaicking to ensure continuity and integrity of all images within the study area. To enhance the accuracy of subsequent analyses, geometric correction was applied to the mosaicked images, eliminating distortions caused by variations in UAV flight altitude and angle. This was followed by radiometric preprocessing to mitigate the impact of sensor sensitivity, solar radiation intensity, and atmospheric conditions on the imagery, ensuring that the images accurately reflected the ground truth. The preprocessed UAV multispectral image information was then imported into ENVI 5.3 software. ENVI is a widely used remote sensing image processing software that supports a variety of data analysis and image processing functions. Within this software, we extracted spectral reflectance, a key metric for measuring the reflection of solar energy by surface objects. To focus on the study area, we clipped corresponding spectral images centered around each experimental plot from the imagery. During the clipping process, special attention was given to exclude areas with soil and film shadows as these could affect the purity of spectral data. Subsequently, regions of interest (ROIs) were defined within each experimental plot, and the average reflectance spectra of the soybean leaf samples were extracted from these areas. This average reflectance spectrum represented the spectral reflectance within the plot, providing us with valuable information about the growth conditions of the soybeans. Ultimately, we obtained spectral reflectance data across different bands, which will be used for further analysis, such as assessing crop health, monitoring vegetation cover changes, or estimating biophysical parameters. Through these detailed spectral data, researchers can gain a deeper understanding of the environmental conditions affecting crop growth, offering a scientific basis for precision agriculture. Figure 3 shows the reflectivity performance of each experimental treatment in each band.

### 2.3. Selection and Construction of Vegetation Index and Texture Features

Crop growth and nutritional status can be effectively reflected by vegetation indices [23]. In this study, based on existing research, ten classic vegetation indices were selected for investigation, with calculation formulas and references provided in Table 1. The texture is a visual feature reflecting homogeneity phenomena in images, indicating the arrangement properties of surface structures with slow or periodic changes. In this paper, ENVI 5.3 software was employed to extract texture features (TFs) based on second-order statistical filtering (co-occurrence measures). Eight TFs were extracted from the near-infrared band: mean (MEA), variance (VAR), homogeneity (HOM), contrast (CON), dissimilarity (DIS), entropy (ENT), second moment (SEM), and correlation (COR). A window size of 7 × 7 and default spatial offset values of 1 were used for texture analysis. To explore the potential applications of texture features in estimating soybean leaf moisture content from UAV multispectral images, randomly combined texture features were extracted in this study. Subsequently, based on previous research experience and formulas, seven types of texture indices (TIs) [24] were constructed, including normalized difference texture index (NDTI), ratio texture index (RTI), difference texture index (DTI), additive texture index (ATI), reciprocal difference texture index (RDTI), and reciprocal additive texture index (RATI). The specific calculation formulas are as follows:(1)$$\mathrm{RTI}=T_{i}/T_{j}$$
(2)$$\mathrm{DTI}=T_{i}-T_{j}$$
(3)$$\mathrm{ATI}=T_{i}+T_{j}$$
(4)$$\mathrm{NDTI}=(T_{i}-T_{j})/(T_{i}+T_{j})$$
(5)$$\mathrm{RDTI}=1/T_{i}-1/T_{j}$$
(6)$$\mathrm{RATI}=1/T_{i}+1/T_{j}$$

**Table 1 plants-13-01498-t001:** Vegetation index and its calculation formula.

Selected Spectra Parameters	Calculation Formula	Reference
Soil-adjusted vegetation index (SAVI)	$(1+0.5)(R_{NIR}-R_{RED}$$)/(R_{NIR}+R_{RED}$+ 0.5)	[25]
Enhanced vegetation index (EVI)	$2.5\times (R_{NIR}-R_{RED})/($ $\left(R_{NIR}+6\times R_{RED}-7.5\times R_{B})+1\right)$	[26]
Modified simple ratio vegetation index (MSR)	$(\frac{R_{NIR}}{R_{RED}}-1$ $)(\frac{R_{NIR}}{R_{RED}}+1{)}^{-0.5}$	[27]
Optimized soil-adjusted vegetation index (OSAVI)	$\left(1+0.16)(R_{NIR}-R_{G})/(R_{NIR}+R_{G}+0.16\right)$	[25]
Renormalized difference vegetation index (RDVI)	(RNIR−RRED/RNIR+RRED)^0.5	[25]
Modified soil-adjusted vegetation index (MSAVI)	$0.5((2R_{NIR}-1$$)+((2R_{NIR}+1{)}^{2}-8(R_{NIR}-R_{RED}{)}^{2}{)}^{0.5}$)	[27]
Atmospheric resistance vegetation index (ARVI)	$(R_{NIR}-{2R}_{RED}+R_{B}$$)/(R_{NIR}+{2R}_{RED}-R_{B}$)	[28]
Green normalized difference vegetation index (GNDVI)	$R_{NIR}-R_{G}/R_{NIR}+R_{G}$	[26]
Meris terrestrial chlorophyll index (MTCI)	$R_{NIR}-R_{RE}/R_{RE}-R_{RED}$	[29]
Chlorophyll index (CI)	$R_{NIR}/R_{RE}-1$	[30]

Note: R_RED, R_G, R_B, R_NIR, and R_RE represent the reflectance of the red, green, blue, near-infrared, and red-edge bands, respectively.

### 2.4. Sample Set Partitioning, Model Methods, and Model Evaluation

During the soybean flowering stage, a total of 54 valid samples were collected. Two-thirds of the samples were randomly selected as the training set, while the remaining one-third was reserved as the validation set. Figure 4 presents the sample counts and statistical characteristics of both the training and validation sets.

First, the correlation between vegetation indices, texture features, and soybean leaf moisture content was analyzed. Parameters significantly correlated with soybean leaf moisture content (*p* < 0.05) were selected as input variables for the model. These include combination 1, vegetation indices; combination 2, texture features; combination 3, texture indices extracted from random combinations; and combination 4, vegetation indices, texture features, and randomly extracted texture indices combined. Subsequently, ELM, XGBoost, and BPNN were employed to model the leaf moisture content. Detailed descriptions of these machine learning models can be found in references [7,31]. 

For the ELM model, a sigmoid function was utilized, and parameters ${\left(a_{i},b_{i}\right)}_{i=1}^{L}$ for the hidden layer were randomly generated within the range [−1, 1]. The number of hidden layer nodes was set to 1000 [7], and the number of neurons started at 15, incrementing by 15 until reaching 120. Each model was run 50 times to select the optimal training result, and the final number of neurons was determined to be 60. 

For the XGBoost algorithm, the optimal parameters were refined through a grid search, setting 100 weak learners (n_estimators), a learning rate of 0.03, and a maximum tree depth (max_depth) of 5 [31]. 

In BPNN, the transfer function for the hidden layer was set as TANSIG, and the Levenberg–Marquardt algorithm based on numerical optimization theory (Train-LM) was used as the network training function. After multiple training iterations, the number of neurons in the middle layer was determined to be 15 [31]. Figure 5 shows the process of UAV multispectral data processing, the acquisition of vegetation index and texture features, and the construction process of soybean leaf moisture content model.

**Figure 4 plants-13-01498-f004:**
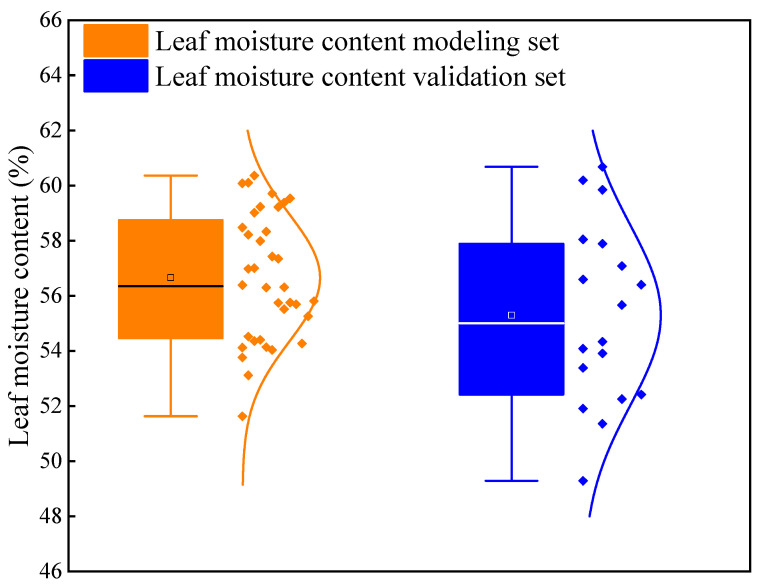
Descriptive statistics of soybean leaf moisture content. The horizontal line in the box line diagram represents the median, and the white box represents the average value.

To validate the model’s prediction accuracy and capability, this study selected three evaluation metrics: the coefficient of determination (R^2^), root mean square error (RMSE), and mean relative error (MRE). These metrics were used to assess the model’s precision [32]. A higher R^2^ value closer to 1 and lower RMSE and MRE values closer to 0 indicated better model fitting. The formulas for these metrics were as follows:(7)$$R^{2}=\frac{\sum_{\dot{i}=1}^{n}{\left({\hat{y}}_{i}-\bar{y}\right)}^{2}}{\sum_{i=1}^{n}{\left(y_{i}-\bar{y}\right)}^{2}}$$
(8)$$RMSE=\sqrt{\frac{\sum_{i=1}^{n}{\left({\hat{y}}_{i}-y_{i}\right)}^{2}}{n}}$$
(9)$$MRE=\frac{1}{n}\sum_{i=1}^{n}\frac{\left|{\hat{y}}_{i}-y_{i}\right|}{y_{i}}\times 100\%$$

**Figure 5 plants-13-01498-f005:**
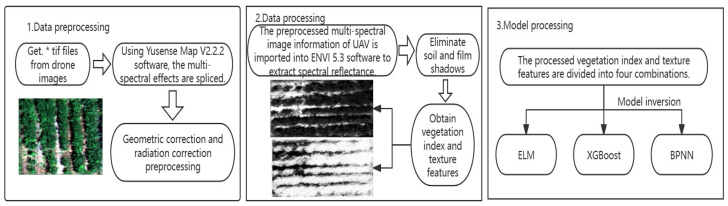
The process of UAV multispectral data processing, the acquisition of vegetation index and texture features, and the construction process of the soybean leaf moisture content model. * on behalf of each small piece of UAV image stitching.

## 3. Results and Analysis

### 3.1. Correlation Analysis between Vegetation Index, Texture Features, and Leaf Moisture Content

The correlation analysis between the vegetation indices and the soybean leaf moisture content is presented in Table 2, the correlation analysis between the texture feature and the soybean leaf moisture content is presented in Table 3. The results indicated that the majority of vegetation indices and texture features exhibited significant correlations with the soybean leaf moisture content (*p* < 0.05). Among these, the vegetation index with the highest correlation coefficient was MSR, with a value of 0.649. Additionally, the texture feature with the highest correlation coefficient to leaf moisture content was the mean in band 2, with a coefficient of 0.644. Subsequently, an analysis of randomly extracted texture indices was conducted (Table 4 and Figure 6). It was found that the randomly extracted texture indices, after screening, also demonstrated significant correlations with soybean leaf moisture content (*p* < 0.05). Among these, RDTI stood out as the combination with the highest correlation coefficient to leaf moisture content, with a coefficient of 0.683. The texture combination comprised Variance1 and Correlation5.

From this, we selected four combinations to serve as input for the model: combination 1 (SAVI, EVI, MSR, OSAVI, RDVI, MSAVI, and ARVI), combination 2 (Mean1, Variance1, Homogeneity1, Contrast1, Dissimilarity1, Entropy1, Second Moment1, Correlation1, Mean2, Variance2, Homogeneity2, Contrast2, Second Moment2, Correlation2, Mean3, Variance3, Homogeneity3, Contrast3, Second Moment3, Correlation3, Mean4, Variance4, Contrast4, Second Moment4, Correlation4, Variance5, Homogeneity5, Contrast5, Dissimilarity5, Second Moment5, Correlation5, Variance6, Homogeneity6, Contrast6, Dissimilarity6, Second Moment6, and Correlation6), combination 3 (RTI, DTI, ATI, NDTI, RATI, and RDTI), combination 4 (SAVI, EVI, MSR, OSAVI, RDVI, MSAVI, ARVI, Mean1, Variance1, Homogeneity1, Contrast1, Dissimilarity1, Entropy1, Second Moment1, Correlation1, Mean2, Variance2, Homogeneity2, Contrast2, Second Moment2, Correlation2, Mean3, Variance3, Homogeneity3, Contrast3, Second Moment3, Correlation3, Mean4, Variance4, Contrast4, Second Moment4, Correlation4, Variance5, Homogeneity5, Contrast5, Dissimilarity5, Second Moment5, Correlation5, Variance6, Homogeneity6, Contrast6, Dissimilarity6, Second Moment6, Correlation6, RTI, DTI, ATI, NDTI, RATI, and RDTI).

**Table 3 plants-13-01498-t003:** Texture features and calculation results of correlation coefficient with soybean leaf moisture content (* significant at *p* < 0.05).

Texture Features	Correlation Coefficients					
	Band 1	Band 2	Band 3	Band 4	Band 5	Band 6
Mean	0.597 *	0.644 *	0.618 *	0.606 *	0.249	0.259
Variance	0.578 *	0.559 *	0.398 *	0.513 *	0.485 *	0.505 *
Homogeneity	0.554 *	0.328 *	0.268 *	0.164	0.429 *	0.424 *
Contrast	0.585 *	0.581 *	0.439 *	0.531 *	0.492 *	0.506 *
Dissimilarity	0.581 *	0.223	0.046	0.170	0.505 *	0.523 *
Entropy	0.389 *	0.210	0.229	0.138	0.251	0.248
Second moment	0.457 *	0.285 *	0.399 *	0.358 *	0.398 *	0.396 *
Correlation	0.471 *	0.593 *	0.599 *	0.588 *	0.667 *	0.654 *

**Table 4 plants-13-01498-t004:** The calculation results of texture index extracted by random combination and correlation coefficient with soybean leaf moisture content (* significant at *p* < 0.05).

Texture Features Extracted by Random Combination	Maximum Correlation Coefficient	
	Correlation Coefficient	Texture Feature Combination
RTI	0.663 *	Homogeneity3, Mean2
DTI	0.645 *	Dissimilarity1, Mean2
ATI	0.653 *	Mean2, Second Moment3
NDTI	0.647 *	Correlation5, Mean4
RATI	0.670 *	Variance5, Correlation5
RDTI	0.683 *	Variance1, Correlation5

**Figure 6 plants-13-01498-f006:**
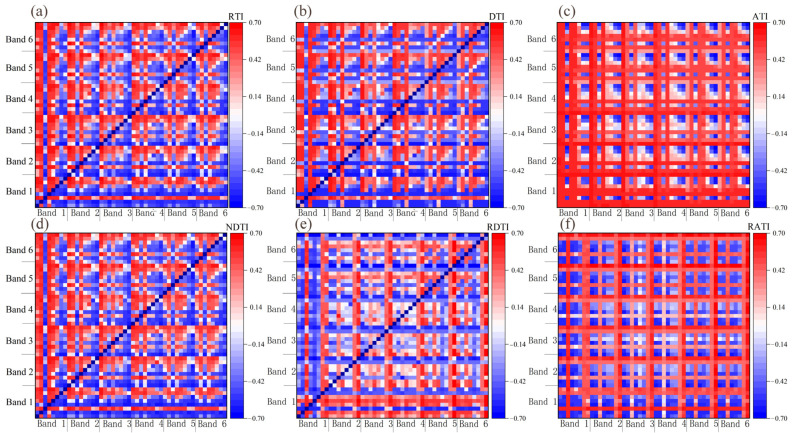
The correlation coefficients between the moisture content and the texture index of soybean leaves were (**a**) RTI, (**b**) DTI, (**c**) ATI, (**d**) NDTI, (**e**) RDTI, and (**f**) RATI. Any point in the figure represents the correlation coefficient between the texture index and the moisture content of soybean leaves. The texture index is calculated by the two texture eigenvalues corresponding to the horizontal and vertical coordinates of the point. Band 1 in the image consists of the following parameters from start to finish: Mean1, Variance1, Homogeneity1, Contrast1, Dissimilarity1, Entropy1, Second Moment1, and Correlation1; band 2 consists of Mean2, Variance2, Homogeneity2, Contrast2, Dissimilarity2, Entropy2, Second Moment2, and Correlation2; band 3 consists of Mean3, Variance3, Homogeneity3, Contrast3, Dissimilarity3, Entropy3, Second Moment3, and Correlation3; band 4 consists of Mean4, Variance4, Homogeneity4, Contrast4, Dissimilarity4, Entropy4, Second Moment4, and Correlation4; band 5 consists of Mean5, Variance5, Homogeneity5, Contrast5, Dissimilarity5, Entropy5, Second Moment5, and Correlation5; and band 6 consists of Mean6, Variance6, Homogeneity6, Contrast6, Dissimilarity6, Entropy6, Second Moment6, and Correlation6.

### 3.2. Construction of a Monitoring Model for Soybean Leaf Moisture Content

The four selected combinations from Section 3.1 were utilized as inputs for modeling using ELM, XGBoost, and BPNN. The model results are illustrated in Figure 7. The findings revealed that when the machine learning models were consistent, the combination of vegetation indices and texture features (combination 4) yielded the highest estimation accuracy for soybean leaf moisture content. This was evidenced by the highest R^2^, and the lowest RMSE and MRE values in the validation set. Furthermore, among models with the same input combinations, XGBoost demonstrated the optimal capability for monitoring the soybean leaf moisture content.

In summary, in this study, the combination of vegetation indices and texture features (combination 4) as input, combined with the XGBoost model, achieved the highest level of soybean leaf moisture content monitoring. The estimated R^2^ for the validation set was 0.816, with an RMSE of 1.404 and an MRE of 1.934%.

**Figure 7 plants-13-01498-f007:**
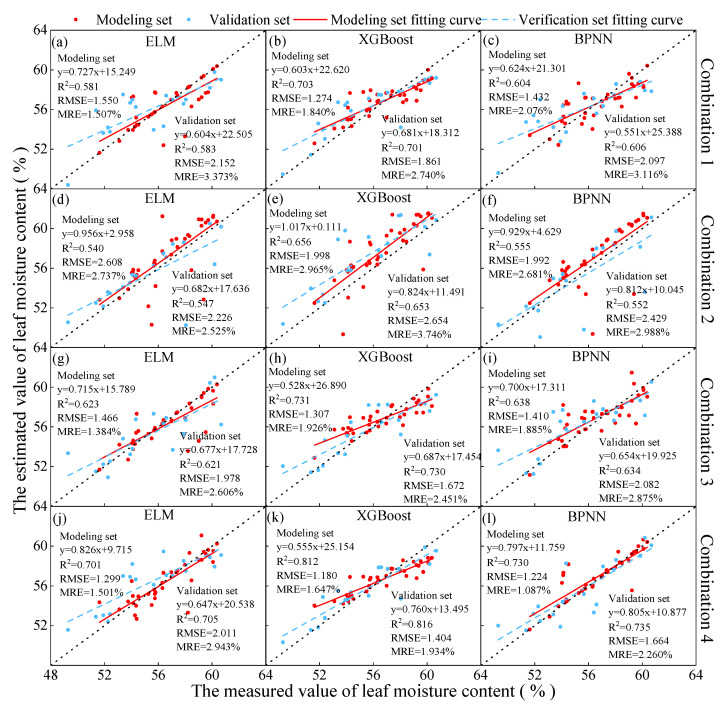
The prediction results of soybean leaf moisture content based on ELM, XGBoost, and BPNN. The red dots in the figure are the modeling set, and the blue dots are the verification set. The predicted results of the soybean leaf moisture content inversion models using different input variables and modeling methods are presented for both the modeling and validation datasets. (**a**−**c**) The prediction models constructed using combination 1 with ELM, XGBoost, and BPNN as the methods. (**d**−**f**) The prediction models constructed using combination 2 with ELM, XGBoost, and BPNN as the methods. (**g**−**i**) The prediction models constructed using combination 3 with ELM, XGBoost, and BPNN as the methods. (**j**−**l**) The prediction models constructed using combination 4 with ELM, XGBoost, and BPNN as the methods.

## 4. Discussion

Water is a crucial element for photosynthesis and nutrient transport in plants. The water content in plant leaves directly influences the plant’s growth process [5]. Therefore, a profound understanding of variations in the leaf moisture content is essential for grasping the plant’s growth status, assessing its water condition, and effectively managing plant water resources. This awareness has garnered widespread attention in fields such as agriculture, forestry, and horticulture [6]. By analyzing vegetation indices and texture features in unmanned aerial vehicle multispectral data, the crop water status can be inferred more accurately, providing a scientific basis for field management and irrigation decisions [12].

In this study, to ensure the accuracy and reliability of experimental outcomes, a stringent control variable approach was adopted. Apart from water management, all other agricultural practices were kept uniform, including but not limited to soil fertility, planting density, and pest control. By doing so, we were able to eliminate the interference of these potential factors on the experimental results, ensuring that the observed outcomes could be attributed to the varying treatments. This research aimed to elucidate the direct impact of water management on soybean growth and yield by precisely controlling all management measures except for water, thereby offering a scientific basis for sustainable agricultural practices.

This study found that the correlation coefficient between vegetation indices and leaf moisture content was generally higher than that of texture features. This is because texture features typically reflect the spatial distribution and structural characteristics of vegetation (such as density, height, coverage, etc.). However, when measuring canopy texture features, the shadowing effect may affect the sensor’s observations, leading to less accurate extraction of canopy texture features [24]. In contrast, vegetation indices are more capable of comprehensively reflecting the growth status of vegetation; therefore, they may be more reliable in terms of correlation with leaf moisture content [33].

The study also found that when using the same machine learning model for modeling, the accuracy of the combined model using combination 4 for leaf moisture content monitoring was higher than combinations 1, 2, and 3. This may be because combination 4, which included vegetation indices, texture features, and texture indices, provided richer and more diverse information. Compared with using vegetation indices or texture features alone, the combination of these two in combination 4 could offer a more comprehensive feature description. Vegetation indices typically reflect the growth status and photosynthetic activity of vegetation [34], while texture features provide information about the structure and spatial distribution of vegetation canopies [35]. By combining these two features, the model could more accurately capture the complex relationship between leaf moisture content and vegetation growth status. Additionally, vegetation indices and texture features often have different sensitivities and feature expression capabilities. That is, vegetation indices may be more suitable for reflecting vegetation growth status, while texture features can better describe the spatial distribution and structural characteristics of vegetation canopies [36]. Therefore, combining these two features can complement each other’s shortcomings and improve the model’s accuracy in estimating the leaf moisture content.

In the process of constructing leaf moisture content models, when the input combinations were the same, it was found that the XGBoost model had higher accuracy compared with the SVM and BPNN models. This may be because XGBoost performed well in handling nonlinear relationships and complex data patterns, enabling better fitting of the complex relationship between leaf moisture content and input features [31]. In contrast, ELM and BPNN models may be less flexible in handling high-dimensional, nonlinear data, resulting in relatively lower accuracy [7,31]. Additionally, XGBoost improved model generalization by optimizing the loss function, which enabled it to perform well beyond the training dataset. This means that the XGBoost model could better adapt to new datasets and exhibit more stable performance on the test set, thereby enhancing model accuracy [37]. Finally, in terms of feature selection and processing, the XGBoost model had unique advantages [22]. It could automatically select the most important features and had good capabilities in handling missing values and outliers, helping to reduce the model’s sensitivity to noise and unnecessary features and thereby improving model accuracy [21].

Currently, monitoring the leaf moisture content based on remote sensing data and machine learning models still has certain limitations. Despite using various feature combinations and machine learning models for modeling, there still exists a certain degree of error and uncertainty. First, vegetation water status is influenced by multiple factors, including climatic conditions, soil types, vegetation types, etc., and existing models may not fully consider the complex relationships among these factors. Additionally, current research mainly focuses on monitoring and evaluating vegetation water status, while challenges remain in translating these monitoring results into effective agricultural management and irrigation decisions in practical applications. Further research and development are needed to establish intelligent agricultural decision support systems based on monitoring results, integrating multiple data sources such as meteorological data, soil information, etc., to achieve precise management and optimal utilization of farmland water resources.

To address these issues, future research will consider in-depth exploration of the water requirements and response patterns of different vegetation types at different growth stages, optimizing vegetation water monitoring models to improve the accuracy and reliability of vegetation water status assessment. Additionally, combining meteorological data, soil information, and other multiple data sources, conducting correlation analyses between vegetation water content and factors such as climatic conditions and soil types, will deepen the understanding of influencing factors and variation patterns of vegetation water status.

## 5. Conclusions

This study employed plot experiments and multispectral data obtained from drones, combined with vegetation indices and texture features, and utilized three machine learning models, extreme learning machine (ELM), extreme gradient boosting tree (XGBoost), and back propagation neural network (BPNN), to estimate soybean leaf moisture content. The results indicated that most vegetation indices and texture features were significantly correlated with the soybean leaf moisture content (*p* < 0.05). Among them, the vegetation index with the highest correlation coefficient was MSR, at 0.649, while the texture feature with the highest correlation coefficient with leaf moisture content was the mean in band 2, at 0.644. All texture indices were significantly correlated with the soybean leaf moisture content (*p* < 0.05), with RATI being the randomly combined texture feature with the highest correlation coefficient, at 0.683. The texture combination was Variance1 and Correlation5, and the prediction model’s fitting accuracy for leaf moisture content was ranked as follows: XGBoost > BPNN > ELM. Furthermore, using the XGBoost model, combination 4 (vegetation indices, texture features, and randomly combined texture features) provided the best monitoring effect for leaf moisture content, with an R^2^ of 0.816, RMSE of 1.404, and MRE of 1.934% on the model validation set. These results provide important references for establishing a nondestructive, rapid, and efficient model for monitoring crop leaf moisture content.

In the research on three machine learning models based on vegetation indices and texture features, there are still some issues to be addressed. For example, this study, along with the majority of researchers, primarily focused on a single growth period of a single plant species as the experimental subject. The feasibility of applying these research findings to the entire growth period of vegetation requires further investigation. Therefore, achieving a higher level of universality and accuracy in simulating leaf moisture content for both individual growth periods and the entire growth period of most plants still requires further research and practical exploration.

## Figures and Tables

**Figure 1 plants-13-01498-f001:**
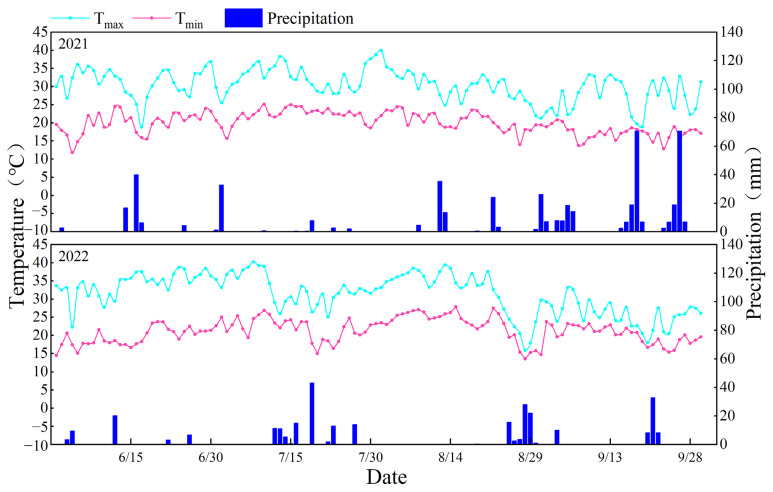
The daily temperature and precipitation of soybean growing season at the Yangling experimental station in China in 2021 and 2022.

**Figure 2 plants-13-01498-f002:**
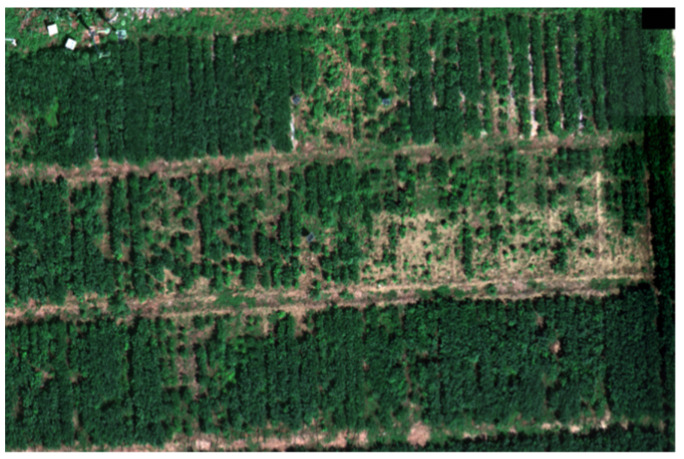
UAV photo of soybean plots in this experimental area.

**Figure 3 plants-13-01498-f003:**
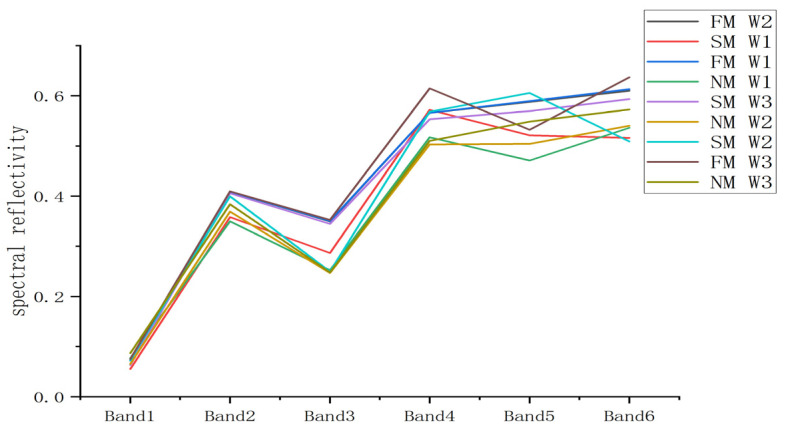
Spectral reflectance of soybean under different field treatments in each band.

**Table 2 plants-13-01498-t002:** The calculation results of vegetation index and correlation coefficient with soybean leaf moisture content (* significant at *p* < 0.05).

Vegetation Index	Correlation Coefficient
Soil-adjusted vegetation index (SAVI)	0.511 *
Enhanced vegetation index (EVI)	0.517 *
Modified simple ratio vegetation index (MSR)	0.649 *
Optimized soil-adjusted vegetation index (OSAVI)	0.636 *
Renormalized difference vegetation index (RDVI)	0.506 *
Modified soil-adjusted vegetation index (MSAVI)	0.619 *
Atmospheric resistance vegetation index (ARVI)	0.643 *
Green normalized difference vegetation index (GNDVI)	0.189
Meris terrestrial chlorophyll index (MTCI)	0.249
Chlorophyll index (CI)	0.171

## Data Availability

Data are contained within the article.

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
