# Peer review of "Soybean (Glycine max L.) Leaf Moisture Estimation Based on Multisource Unmanned Aerial Vehicle Image Feature Fusion"

_plants, 2024, doi:10.3390/plants13111498_

Round 1

Reviewer 1 Report

Comments and Suggestions for Authors

This manuscript (plants-3021386) demonstrates that efficient monitoring of crop leaf moisture is crucial for effective irrigation and maximizing yield. Using UAV multispectral technology over two years, the study collected data and images of soybean leaf moisture. By analyzing vegetation indices, canopy texture features, and randomly extracted texture indices, significant correlations with leaf moisture were found. Extreme learning machine, eXtreme gradient boosting, and Back Propagation Neural Network models were used for moisture estimation. The XGBoost model, using a combination of indices and features, achieved the highest accuracy with an R² of 0.816, RMSE of 1.404, and MRE of 1.934%, supporting UAV multispectral monitoring for rapid crop growth assessment.

The sections of the manuscript are appropriate, precise, and informative. The results are interesting and suitable. Minor corrections need to be made in the captions.

Keywords in alphabetical order;

L148. Analytic balance;

Figure 2. It is not possible to identify what is written on the axes.

Figure 3. The captions need to be better described, including describing the colored points. This should be done in the same way in the other captions of the tables and figures. Include the sample number, as it is difficult to quote or identify appropriately.

Questions and suggestions to the authors:

Why didn’t the authors add images of the drone flight and the image records?

The authors could show the obtained images; I suggest a flowchart demonstrating the applied process steps.

What were the conditions for obtaining the spectral data? Is it possible to add the multispectral curves?

Comments on the Quality of English Language

Language corrections need to be made, improving fluency, grammar, and spelling.

Author Response

Responses to Reviewer #1 (Manuscript ID: Plants- 3021386)

This manuscript (plants-3021386) demonstrates that efficient monitoring of crop leaf moisture is crucial for effective irrigation and maximizing yield. Using UAV multispectral technology over two years, the study collected data and images of soybean leaf moisture. By analyzing vegetation indices, canopy texture features, and randomly extracted texture indices, significant correlations with leaf moisture were found. Extreme learning machine, eXtreme gradient boosting, and Back Propagation Neural Network models were used for moisture estimation. The XGBoost model, using a combination of indices and features, achieved the highest accuracy with an R² of 0.816, RMSE of 1.404, and MRE of 1.934%, supporting UAV multispectral monitoring for rapid crop growth assessment. The sections of the manuscript are appropriate, precise, and informative. The results are interesting and suitable. Minor corrections need to be made in the captions.

Thank you for your careful review and positive comments. We have now incorporated the reviewers comments and suggestions in preparation of the revised manuscript. The modified part is marked in red in the manuscript.

  1. Keywords in alphabetical order.

Response: Thank you for your pointing out, we have revised according to the requirements.

Line35-36: Keywords; Leaf moisture content; Multispectral; Soil moisture content; Soybean; Texture features; Vegetation indices

  1. Analytic balance;

Response: Thank you for pointing out that we have made changes according to your suggestions. Maybe it is because of the writing problem, we will pay attention next time.

Line158:  using an analytic balance

  1. Figure 2. It is not possible to identify what is written on the axes.

Response: Thank you for pointing this out. I've improved Figure 6 (Figure 2) to recognize what's on the axis and leave a detailed description in the comments below.

Fig 6. The correlation coefficients between moisture content and texture index of soybean leaves were (a) RTI, (b) DTI, (c) ATI, (d) NDTI, (e) RDTI, (f) RATI. Any point in the figure represents the correlation coefficient between the texture index and the moisture content of soybean leaves. The texture index is calculated by the two texture eigenvalues corresponding to the horizontal and vertical coordinates of the point. Band 1 in the image consists of the following parameters from start to finish: Mean1, Variance1, Homogeneity1, Contrast1, Dissimilarity1, Entropy1, Second Moment1, Correlation1; Band 2 consists of: Mean2, Variance2, Homogeneity2, Contrast2, Dissimilarity2, Entropy2, Second Moment2, Correlation2; Band 3 consists of: Mean3, Variance3, Homogeneity3, Contrast3, Dissimilarity3, Entropy3, Second Moment3, Correlation3; Band 4 consists of: Mean4, Variance4, Homogeneity4, Contrast4, Dissimilarity4, Entropy4, Second Moment4, Correlation4; Band 5 consists of: Mean5, Variance5, Homogeneity5, Contrast5, Dissimilarity5, Entropy5, Second Moment5, Correlation5; Band 6 consists of: Mean6, Variance6, Homogeneity6, Contrast6, Dissimilarity6, Entropy6, Second Moment6, Correlation6.

  1. Figure 3. The captions need to be better described, including describing the colored points. This should be done in the same way in the other captions of the tables and figures. Include the sample number, as it is difficult to quote or identify appropriately.

Response: Thank you for pointing this out. I have added a description of the title and color points in the comments below Figure 7 (Figure 3), and the picture number is also described in detail.

Fig.7 The Prediction results of soybean leaf moisture content based on ELM, XGBoost and BPNN. The red dots in the figure are the modeling set, and the blue dots are the verification set. The predicted results of the soybean leaf moisture content inversion models using different input variables and modeling methods are presented for both the modeling and validation datasets. (a-c) depict the prediction models constructed using Combination 1 with ELM, XGBoost, and BPNN as the methods. (d-f) illustrate the prediction models constructed using Combination 2 with ELM, XGBoost, and BPNN as the methods. (g-i) demonstrate the prediction models constructed using Combination 3 with ELM, XGBoost, and BPNN as the methods. (j-l) display the prediction models constructed using Combination 4 with ELM, XGBoost, and BPNN as the methods.

  1. Why didn’t the authors add images of the drone flight and the image records?The authors could show the obtained images; I suggest a flowchart demonstrating the applied process steps.

Response: Thank you for pointing this out. We added the captured image of the drone in the article, as well as the corresponding flow chart, and displayed the processed drone image in the flow chart.

Fig.2 UAV diagram.

Fig.5 UAV flow chart.

  • What were the conditions for obtaining the spectral data? Is it possible to add the multispectral curves?

Response: Thank you very much for your comments. We have supplemented the article according to your suggestions, improved the conditions for obtaining spectral data, and added a multi-spectral line chart.

Line 166-191: In scientific research, precise handling and analysis of remote sensing data are crucial. In this study, we employed the Yusense Map V2.2.2 software to process multispectral imagery collected by unmanned aerial vehicles (UAVs). Initially, the software was used for image mosaicking to ensure continuity and integrity of all images within the study area. To enhance the accuracy of subsequent analyses, geometric correction was applied to the mosaicked images, eliminating distortions caused by variations in UAV flight altitude and angle. This was followed by radiometric preprocessing to mitigate the impact of sensor sensitivity, solar radiation intensity, and atmospheric conditions on the imagery, ensuring that the images accurately reflected the ground truth. The preprocessed UAV multispectral image information was then imported into ENVI 5.3 software. ENVI is a widely used remote sensing image processing software that supports a variety of data analysis and image processing functions. Within this software, we extracted spectral reflectance, a key metric for measuring the reflection of solar energy by surface objects. To focus on the study area, we clipped corresponding spectral images centered around each experimental plot from the imagery. During the clipping process, special attention was given to exclude areas with soil and film shadows, as these could affect the purity of spectral data. Subsequently, regions of interest (ROIs) were defined within each experimental plot, and average reflectance spectra of soybean leaf samples were extracted from these areas. This average reflectance spectrum represented the spectral reflectance within the plot, providing us with valuable information about the growth conditions of the soybeans. Ultimately, we obtained spectral reflectance data across different bands, which will be used for further analysis, such as assessing crop health, monitoring vegetation cover changes, or estimating biophysical parameters. Through these detailed spectral data, researchers can gain a deeper understanding of the environmental conditions affecting crop growth, offering a scientific basis for precision agriculture.

Fig.3 multi-spectral

Reviewer 2 Report

Comments and Suggestions for Authors

The manuscript entitled „Soybean Leaf Moisture Estimation Based on Multi-Source Unmanned Aerial Vehicle Image Feature Fusion” presents interesting study on UAV application for evaluation of leaf moisture.

The manuscript is not formatted acoording guidelines for authors. For example letters next to the authors shuld be written as superscripts. Font format in the tables is other than recommended. Please follow the guidelines for the authors.

Line 115: Please format the „-1” properly, i.e.as a superscript.

Line 101: It would be better if you provide weather conditio for the period of the study not veraged precipitation.

Line 133: What do you mean as a „pixel resolution of 1.2×106”? It is not clear. Please explain.

It is not clear if the statistical analyses were performed at plot level or smaller (individual plants). Please provide more details about plant sampling for evaluation of leaf moisture.

There are no information at whch crop growth stage the analyses were performed. Radiation reflectance is probably very different depending on many factors, including growth stage, crop management, cultivar. Did you consider these factors?

Charts in the Fig. 2 are very small and not possible to read. Please increase the size of these charts to be clear.

The conclusions are very general. Please be more specific. The study was conducted in certain conditions (including growth stage, crop management, cultivar ) and probably the results are valid only for these conditions? Please provide limitation of the study.

Author Response

Responses to Reviewer #2 (Manuscript ID: Plants- 3021386)

Thank you for addressing all my comments.

Thank you for your careful review and positive comments. We have now incorporated the reviewer’s comments and suggestions in preparation of the revised manuscript. The modified part is marked in red in the manuscript.

  1. The manuscript is not formatted according guidelines for authors. For example letters next to the authors should be written as superscripts. Font format in the tables is other than recommended. Please follow the guidelines for the authors.

Response: Thank you for pointing this out. I have revised according to the requirements.

Wanli Yanga,b, Zhijun Lia,b,*, Guofu Chena,b, Shihao Cuia,b, Yue Wua,b, Xiaochi Liua,b, Wen Menga,b, Yucheng Liua,b, Jinyao Hea,b, Danmao Liua,b, Yifan Zhoua,b, Zijun Tanga,b, Youzhen Xianga,b, Fucang Zhanga,b

  1. Line 115: Please format the „-1” properly, i.e.as a superscript.

Response: Thank you for pointing out that we have made changes. This is caused by our carelessness, we will pay attention next time.

Line123-135: Before sowing, each plot received phosphorus and potassium fertilizers at a rate of 30 kg ha-1 and nitrogen fertilizer at a rate of 120 kg ha-1. The nitrogen fertilizer used in the experiment was urea (46% N), the phosphorus fertilizer was calcium superphosphate (16% P2O5), and the potassium fertilizer was potassium chloride (62% K2O).

For the FM treatment, a ridge (50 cm wide, 30 cm high) was used, and seeds were sown in furrows covered by the ridge. The ridge-furrow ratio was 1:1. Before sowing, two rows of soybeans were planted at the bottom of each ridge. The straw mulch rate was 9000 kg ha-1, and wheat straw was used to cover the soil within 7 days after sowing. The planting density of soybeans was 300,000 plants ha-1, with row spacing of 50 centimeters and plant spacing of 10 centimeters. Soybeans were sown on June 18, 2021, and June 10, 2022, and harvested on September 30, 2021, and September 20, 2022, respectively.

  1. Line 101: It would be better if you provide weather conditio for the period of the study not veraged precipitation.

Response: Thank you for pointing this out. I have increased the weather conditions in the experimental site according to your requirements.

Line102-110: Daily temperature and rainfall data for the two growing seasons were carefully recorded by an automatic weather station located within the experimental fields, as shown in Figure 1. During the period from June to October 2021, the annual average maximum and minimum temperatures were 30.3°C and 20.0°C, respectively, while in 2022, they were 31.3°C and 21.2°C, respectively. The precipitation during the soybean growing seasons in 2021 and 2022 was 432.6 millimeters (from June 18, 2021, to September 30, 2021) and 279.5 millimeters (from June 10, 2022, to September 20, 2022), respectively. For basic terrain and meteorological information of the experimental site, please refer to reference [1].

Fig.1 The local weather conditions during the study period.

  1. Line 133: What do you mean as a „pixel resolution of 1.2×106”? It is not clear. Please explain.

Response: Thank you for pointing this out. We have explained in the original text that 1.2×106” is the camera pixel, and the correct writing should be 1.2×106”. We have made corrections and explanations.

Line 144: with a pixel resolution of 1.2 × 106.

  1. It is not clear if the statistical analyses were performed at plot level or smaller (individual plants). Please provide more details about plant sampling for evaluation of leaf moisture.

Response: Thank you for pointing this out. We have added the corresponding explanation in the article. We used Five average growing soybeans to represent the leaf moisture content of the entire region with their leaf moisture content.

Line155-163: Simultaneous to the collection of multispectral information by drones, soybean leaf moisture content was determined using the drying method. Five average growing soybeans plants were selected from each plot. Fresh leaves were harvested from various directions and heights of each plant, totaling 100 grams, using an analytic balance. These leaves were then placed in parchment bags, labeled, and subjected to dehydration in a drying oven at 105°C for 0.5 hours, followed by drying at 80°C until a constant mass was achieved. The dry mass, after subtracting the mass of the parchment bags, represented the moisture content. The average moisture content of the five soybean plants was considered indicative of the entire plot's soybean leaf moisture content

  1. There are no information at whch crop growth stage the analyses were performed. Radiation reflectance is probably very different depending on many factors, including growth stage, crop management, cultivar. Did you consider these factors?

Response: Thank you for pointing this out. We have shown in the article that it is analyzed in the flowering period of soybean. Radiation reflectance may vary greatly due to many factors, including growth stage, crop management, and variety. In this study, to ensure the accuracy and reliability of experimental outcomes, a stringent control variable approach was adopted. Apart from water management, all other agricultural practices were kept uniform, including but not limited to soil fertility, planting density, and pest control. By doing so, we were able to eliminate the interference of these potential factors on the experimental results, ensuring that the observed outcomes could be attributed to the varying treatments. This research aimed to elucidate the direct impact of water management on soybean growth and yield by precisely controlling all management measures except for water, thereby offering a scientific basis for sustainable agricultural practices.

  1. Charts in the Fig. 2 are very small and not possible to read. Please increase the size of these charts to be clear.

Response: Thank you for pointing this out. I've improved Figure 6 (Figure 2) to recognize what's on the axis and leave a detailed description in the comments below.

Fig 6. The correlation coefficients between moisture content and texture index of soybean leaves were (a) RTI, (b) DTI, (c) ATI, (d) NDTI, (e) RDTI, (f) RATI. Any point in the figure represents the correlation coefficient between the texture index and the moisture content of soybean leaves. The texture index is calculated by the two texture eigenvalues corresponding to the horizontal and vertical coordinates of the point. Band 1 in the image consists of the following parameters from start to finish: Mean1, Variance1, Homogeneity1, Contrast1, Dissimilarity1, Entropy1, Second Moment1, Correlation1; Band 2 consists of: Mean2, Variance2, Homogeneity2, Contrast2, Dissimilarity2, Entropy2, Second Moment2, Correlation2; Band 3 consists of: Mean3, Variance3, Homogeneity3, Contrast3, Dissimilarity3, Entropy3, Second Moment3, Correlation3; Band 4 consists of: Mean4, Variance4, Homogeneity4, Contrast4, Dissimilarity4, Entropy4, Second Moment4, Correlation4; Band 5 consists of: Mean5, Variance5, Homogeneity5, Contrast5, Dissimilarity5, Entropy5, Second Moment5, Correlation5; Band 6 consists of: Mean6, Variance6, Homogeneity6, Contrast6, Dissimilarity6, Entropy6, Second Moment6, Correlation6.

  1. The conclusions are very general. Please be more specific. The study was conducted in certain conditions (including growth stage, crop management, cultivar ) and probably the results are valid only for these conditions? Please provide limitation of the study.

Response: Thank you for pointing this out.We have supplemented and explained the end according to your requirements, and increased the limitations of our experiment to facilitate people to have a clear understanding.

Line 414-439: This study employed plot experiments and multispectral data obtained from drones, combined with vegetation indices and texture features, and utilized three machine learning models, Extreme Learning Machine (ELM), Extreme Gradient Boosting Tree (XGBoost), and Back Propagation Neural Network (BPNN), to estimate soybean leaf moisture content. Results indicate that most vegetation indices and texture features are significantly correlated with soybean leaf moisture content (P<0.05). Among them, the vegetation index with the highest correlation coefficient is MSR, at 0.649, while the texture feature with the highest correlation coefficient with leaf moisture content is the mean in Band 2, at 0.644. All texture indices are significantly correlated with soybean leaf moisture content (P<0.05), with RATI being the randomly combined texture feature with the highest correlation coefficient, at 0.683. The texture combination is Variance1 and Correlation5, and the prediction model's fitting accuracy for leaf moisture content is ranked as follows: XGBoost > BPNN > ELM. Furthermore, using the XGBoost model, combination 4 (vegetation indices, texture features, and randomly combined texture features) provides the best monitoring effect for leaf moisture content, with an R2 of 0.816, RMSE of 1.404, and MRE of 1.934% on the model validation set. These results provide important references for establishing a non-destructive, rapid, and efficient model for monitoring crop leaf moisture content.

In the research on three machine learning models based on vegetation indices and texture features, there are still some issues to be addressed. For example, this study, along with the majority of researchers, primarily focuses on a single growth period of a single plant species as the experimental subject. The feasibility of applying these research findings to the entire growth period of vegetation requires further investigation. Therefore, achieving a higher level of universality and accuracy in simulating leaf moisture content for both individual growth periods and the entire growth period of most plants still requires further research and practical exploration.

Round 2

Reviewer 2 Report

Comments and Suggestions for Authors

The manuscript was improved according all my suggestions but formatting of tha manuscript still demands adjusting. Please follow the guidelines for authors. For example affiliations of the authors should be referenced as 1 and 2 not a and b.

Moreover, the titles of the tables and figures should be selfexplanatory, i.e. clear enough without reading all the manuscript. In current version the titles are very short and sould be extended and more specific.

Author Response

Responses to Reviewer  (Manuscript ID: Plants- 3021386)

Thank you for addressing all my comments.

Thank you for your careful review and positive comments. We have now incorporated the reviewers comments and suggestions in preparation of the revised manuscript. The modified part is marked in red in the manuscript.

  • The manuscript was improved according all my suggestions but formatting of tha manuscript still demands adjusting. Please follow the guidelines for authors. For example affiliations of the authors should be referenced as 1 and 2 not a and b.

Response: Thank you for your pointing out, we have revised according to the requirements.

Line 4-5:Wanli Yang1,2, Zhijun Li1,2,*, Guofu Chen1,2, Shihao Cui1,2, Yue Wu1,2, Xiaochi Liu1,2, Wen Meng1,2, Yucheng Liu1,2, Jinyao He1,2, Danmao Liu1,2, Yifan Zhou1,2, Zijun Tang1,2, Youzhen Xiang1,2, Fucang Zhang1,2

  • Moreover, the titles of the tables and figures should be selfexplanatory, i.e. clear enough without reading all the manuscript. In current version the titles are very short and sould be extended and more specific.

Response: Thank you for your pointing out. We have been modified according to your requirements, and the title of some charts has been supplemented and explained to facilitate better reading.

Line 112: Fig.1 The daily temperature and precipitation of soybean growing season in yangling experimental station of China in 2021 and 2022.

Line 195: Fig.2 UAV photo of soybean plots in this experimental area.

Line 198: Fig.3 Spectral reflectance of soybean under different field treatments in each band.

Line 252-253: Fig.4 Descriptive statistics of soybean leaf moisture content. The horizontal line in the box line diagram represents the median, and the white box represents the average value.

Line 263-264: Fig.5 The process of UAV multispectral data processing, the acquisition of vegetation index and texture features, and the construction process of soybean leaf moisture content model.
